# Effectiveness of Self-Affirmation Interventions in Educational Settings: A Meta-Analysis

**DOI:** 10.3390/healthcare12010003

**Published:** 2023-12-19

**Authors:** Carolang Escobar-Soler, Raúl Berrios, Gabriel Peñaloza-Díaz, Carlos Melis-Rivera, Alejandra Caqueo-Urízar, Felipe Ponce-Correa, Jerome Flores

**Affiliations:** 1Programa de Doctorado en Psicología, Universidad de Tarapacá y Universidad Católica del Norte, Arica 1000000, Chile; gabriel.penaloza.diaz@alumnos.uta.cl (G.P.-D.); carlos.melis.rivera@alumnos.uta.cl (C.M.-R.); fponcec@academicos.uta.cl (F.P.-C.); 2Centro de Justicia Educacional (CJE), Pontificia Universidad Católica de Chile, Santiago 7820436, Chile; jflores@uta.cl; 3Departamento de Administración, Facultad de Administración y Economía, Universidad de Santiago de Chile, Santiago 7820436, Chile; raul.berrios@usach.cl; 4Instituto de Alta Investigación, Universidad de Tarapacá, Arica 1000000, Chile; acaqueo@academicos.uta.cl

**Keywords:** self-affirmation, intervention, effects, education, schoolchildren, college students

## Abstract

School and university can be stressful contexts that can become an important source of identity threats when social prejudices or stereotypes come into play. Self-affirmation interventions are key strategies for mitigating the negative consequences of identity threat. This meta-analysis aims to provide an overview of the effectiveness of self-affirmation interventions in educational settings. A peer-reviewed article search was conducted in January 2023. A total of 144 experimental studies that tested the effect of self-affirmation interventions in educational contexts among high school and university students from different social and cultural backgrounds were considered. The average effect of self-affirmation interventions was of low magnitude (*d*_IG_+ = 0.41, z = 16.01, *p* < 0.00), with a 95% confidence interval whose values tended to lie between 0.36 and 0.45 (*SE* = 0.0253). In addition, moderators such as identity threat, participants’ age, and intervention procedure were found. Through a meta-analysis of the impact of self-affirmation interventions in educational contexts, this study suggests that interventions are effective, resulting in a small mean effect size. Thus, self-affirmation interventions can be considered useful, brief, and inexpensive strategies to improve general well-being and performance in educational settings.

## 1. Introduction

Self-affirmation as an intervention strategy aims to protect students’ self-esteem by promoting positive self-evaluations and creating self-schemas that act as authentic emotional and affective support in the face of negative experiences such as learning difficulties or school failure [1]. Another experience that can threaten students’ sense of self is discrimination or segregation (stereotype threat) for belonging to a minority group, whereby they are forced to adopt defensive mechanisms that lead them to make their own values invisible and begin to identify with those desired by others [2,3].

Stereotype threat is a real obstacle to the development of students’ identities (especially at critical stages of the life cycle, such as adolescence). It causes socioaffective reactions oriented toward the activation of defensive mechanisms that lead to a reconceptualization of the self and one’s own identity in order to adapt to an environment they perceive as threatening. In this scenario, self-affirmation succeeds in promoting genuinely positive self-definitions that immunize students against the threat of stereotyping [4].

The effects of self-affirmation drive not only the evaluation of difficult circumstances in the form of hopeful or resilient coping against identity threats, but also a positive and broad valuation of self-concept and self-image, where the individual recognizes him/herself as good and capable according to the characteristics of his/her own social and cultural context [5,6,7]. The central idea proposed by this theory suggests that the affirmation of the self points to the recognition and valuation of a broader and more versatile self-concept capable of drawing on a more significant number of personal resources in the face of identity threats. A broad perspective and resources of the self allows us to avoid focusing our attention on the threat, overestimating its importance, and consequently exacerbating its negative effect. Adopting an optimistic and resilient disposition allows people to feel capable of resolving perceived threats and safeguarding psychological well-being [8,9,10]. It should be pointed out that self-affirmation is not synonymous with self-praise; instead, it helps develop the conviction of being worthy of praise or admiration by others, according to what is desirable and approved in a given context.

Individuals frequently face different types of identity threats; therefore, they have different self-affirmation needs. In this regard, the effects of self-affirmation are usually more pronounced when there is a high perception of threat and when it is essential to defend or protect oneself from it by exerting self-confidence or self-efficacy to reduce or mitigate it [8].

When an individual feels highly threatened, the effects of self-affirmation promote perceptions of self-efficacy through which they begin to recognize more personal resources and, therefore, more confidence to manage and cope with the threat [10,11].

This theory proposes three fundamental principles to explain the positive and persistent long-term effects of self-affirmation: (1) recursion, (2) interaction, and (3) subjective interpretation. The first is related to the recurrence of self-affirmation’s effects, as the perception of achievement or progress becomes an important incentive to reassert oneself and continue improving. The second principle is associated with the contexts in which individuals develop. By trusting their resources and feeling increasingly capable of growing in different areas of their lives, individuals tend to show changes that others can perceive, evaluate, and value positively, triggering the effects of self-affirmation from an intersubjective source. Finally, the third principle is related to the reframing of identity in individuals who change the way they evaluate and perceive themselves in such a way that they manage to feel satisfied and motivated to continue discovering and progressing [12,13,14].

Particularly in educational settings, stressful experiences extend beyond studying, taking exams, and writing essays; they include relationships with classmates, which can become an essential source of identity threats when prejudices or social stereotypes emerge [5,15,16]. Indeed, a study conducted on adolescents of African descent who perceived hostility in their educational environment showed that stress levels were not related to academic aspects, but relational ones, with lower grades observed in this group of students. The study found that interventions based on self-affirmation reduced the perception of threat and predicted better academic performance in a group of students of African descent [17].

In another study, results showed that when Latin American students perceived less threat from their classmates, they performed better academically, with similar grades as North American or European students [10].

Other studies using self-affirmation interventions in an educational context with social segregation have also reported improvements in the sense of belonging to groups and social competencies [7,9].

Belonging to a stigmatized group can jeopardize self-integration and negatively affect the sense of personal adequacy in the educational context, compromising not only self-esteem but also learning and academic performance [18,19,20,21]. Given this scenario, self-affirmation manages to reduce stereotype threats, facilitating students’ adaptation to the school environment and their functioning, safeguarding positive self-evaluations, and favoring progress in their studies despite the hostility they may perceive [22,23].

Some studies have found that self-affirming students successfully cope with experiences of discrimination. Even when they experience different difficulties (e.g., problems with regulating emotions, feelings of inadequacy and inferiority, nonconformity with self-image, or affective distancing from family), adolescents can thrive and obtain better grades [7,10,24,25,26].

Regarding interventions in educational settings, for more than a decade, experimental studies have attempted to examine the effects of self-affirmation, particularly in minority and stigmatized groups. For example, strategies such as expressive writing have been used to promote identity self-affirmation in a diverse sociocultural classroom population with improved academic achievement and well-being, while observing greater effectiveness in groups where psychological threats are intense and imminent [7,8,10,17,24,25,27,28,29,30,31].

Some studies have found minimal or even significant effects of self-affirmation interventions and suggesting these could be explained by individual differences between participants. It cannot be assumed that individual responses to identity threats will be the same, even if social or cultural origins correspond to those of studied minorities or segregated groups. Similarly, the way in which the threat affects the identity of an individual or a group depends, to a large extent, on the social and cultural characteristics of the context. Therefore, it is not plausible to expect these interventions to have a positive impact if it is not certain that these individuals or groups are potentially threatened in their own contexts [32]. In this sense, researchers suggest an exhaustive search and a more precise definition of moderators related to individual differences and methodological aspects that may explain the variation in these interventions across studies. We suggest so in this meta-analysis and mainly focus on the review of moderators related to sample characteristics and on the methodological designs of the studies.

Despite the diversity of theoretical conjectures regarding the conditions that influence the effectiveness of these interventions, there is insufficient empirical evidence to make practical sense of and explain why there is so much heterogeneity in the effects of these interventions across studies whose outcome variables have remained the same.

The present meta-analysis aimed to provide a broader and more general overview of the effectiveness of self-affirmation interventions on psychosocial, academic, and health variables in high school and university students and to provide estimates that may shed light on the methodological conditions that are sufficient and necessary to ensure favorable results, regardless of the outcome variable on which the effects of these interventions can be analyzed.

## 2. Methods

### 2.1. Eligibility Criteria

The included studies shared seminal theoretical principles of self-affirmation interventions, intervention procedures, and methodological delimitation. Similarly, the study populations coincided in terms of participants, forming groups with and without identity threats as the key elements. In line with the aim of this review, the studies generally measured the effectiveness of self-affirmation interventions in educational contexts.

In order to ensure the rigor and relevance of our study selection process, we established the following inclusion criteria: (1) the studies employed an experimental or quasi-experimental design with a nonclinical sample of subjects; (2) the studies utilized a single self-affirmation intervention strategy (studies whose intervention proposal considered more than one strategy were excluded); (3) the study samples consisted of school or university students in diverse educational and sociocultural contexts; (4) from a methodological perspective, the studies reported a random or at least quasi-random assignment of participants to experimental and control groups; (5) the studies manipulated students’ self-affirmation experience through expressive writing on a significant personal value (a technique most commonly used to promote self-affirmation effects in students) and reported the effectiveness of this intervention on academic performance, aspects of psychological well-being, or interpersonal development, offering intra- and intergroup comparisons. Sometimes, the full texts of the identifiable reports were not available; therefore, they were excluded from the analyses.

If studies reported more than one sample, the effect size of each sample was included in the meta-analysis. If studies reported multiple measures of the same construct, the average effect size for that construct within the study was calculated and coded. In cases where the studies reported multiple outcomes, multiple effect sizes were coded. Finally, for the longitudinal studies, the effect sizes of the self-affirmation interventions on the outcome variables were extracted. Specifically, if a study had only one temporal follow-up point (e.g., T2), the effect was extracted at T2. Conversely, if the study included multiple temporal follow-up points, all effect sizes were extracted and the mean was calculated.

### 2.2. Search Strategy

In January 2023, peer-reviewed articles investigating the impact of self-affirmation interventions in educational contexts were retrieved through systematic searches of the Web of Science, ProQuest, APA PSYCNET, and Google Scholar databases. The search for articles was temporally placed between the years 1990 and 2023, using the following keywords: “Self-Affirmation Interventions AND Educational contexts”, “Self-Affirmation Interventions AND Academic contexts”, “Self-Affirmation Interventions AND Children”, “Self-Affirmation Interventions AND Teenagers/Adolescents”, “Self-Affirmation Interventions AND Elementary School”, “Self-Affirmation Interventions AND Classroom”, “Self-Affirmation Interventions AND High School”, “Self-Affirmation Interventions AND Secondary School”, “Self-Affirmation Interventions AND Primary School”, “Self-Affirmation Interventions AND Students”, “Self-Affirmation Interventions AND Minorities”, “Self-Affirmation Interventions AND Stereotype”, “Self-Affirmation Interventions AND University”, “Self-Affirmation Interventions AND College”, “Self-Affirmation Interventions AND Academic”, “Self-Affirmation Interventions AND Undergraduate”, “Self-Affirmation Interventions AND Degree”, and “Self-Affirmation Interventions AND Bachelor”. Further, the reference sections of the most relevant articles were examined and emails were sent to relevant researchers in the field to search for articles that could not be identified through the databases. Finally, no geographical restrictions were applied; however, studies were limited to those published in English or Spanish.

### 2.3. Assessment of the Methodological Quality

The “Quality Assessment Tool for Quantitative Studies”, developed by the Effective Public Health Practice Project (EPHPP), was applied to comprehensively evaluate methodological quality in the selected studies. This tool, tailored for quantitative research, examines various dimensions, including selection bias, study design, confounders, blinding, data collection methods, and withdrawals and dropouts.

We examined participant representation for *selection bias*, ensuring alignment with the target population. The analysis of *study design* was aimed at appropriateness and robustness. Strategies to address *confounders* were assessed, and *blinding* procedures were thoroughly reviewed to gauge participant and assessor blinding adequacy. Additionally, the tool systematically explored *data collection methods*, emphasizing validity and reliability assessment. Lastly, the evaluation of *withdrawals and dropouts* focused on the studies’ capability to acknowledge and explain reasons for incomplete participant data.

Each component was rated as strong, moderate, or weak, providing an overall score for the methodological quality of each study. Independent assessments were conducted by two raters, and any discrepancies were resolved through consultation with the first author.

### 2.4. Data Extraction and Data Items

A wide range of study characteristics were systematically coded, considering their potential influence on the variability of effect sizes. Study characteristics included (a) year of publication and (b) country from which data were collected. Sample sociodemographic data encompassed (a) the mean or age range of the participants, (b) cultural origin, (c) economic status, (d) sample size, and (e) dominant racial group. Various methodological characteristics were also recorded. These included (a) type of execution modality (face-to-face or virtual), (b) rigor of randomization, (c) dependent variable type, (d) self-affirmation intervention style, (e) placebo style, (f) timing of the intervention (before, during, or after stressful academic events), (g) presentation format of the intervention, and (h) teacher role. The first and third authors independently reviewed and systematically coded all eligible studies. Throughout the coding phase, the second author continuously checked the quality of the codes to identify divergent coding. Discrepancies were resolved by consensus.

Moderators were coded according to the sociodemographic characteristics of the sample and the methodological design (see Table 1). These moderators are in line with those suggested in the literature, pointing to features of the delivery of activities, individual characteristics of the participating students, and aspects of the social context [32,33]. Methodological procedural aspects were used as moderating variables to address the need to delimit sufficient preconditions for interventions in educational contexts [13].

### 2.5. Calculation of Effect Sizes

Effect sizes were calculated according to the statistical indicators of change reported in each study for the variations observed in the experimental and control groups owing to the self-affirmation intervention.

Because all studies used a design with independent groups (experimental and control), it was not necessary to consider a transformation to alternative metrics to guarantee the compatibility of effect size estimates [34], which were calculated according to a standard metric of independent groups (*d*_IG_).

### 2.6. Meta-Analysis Strategy

Calculations were performed using the Statistical Package for the Social Sciences (SPSS) version 25.0. A random effects model was selected because of the variability in effect sizes depending on the characteristics of the samples and the methodological design of the studies.

The restricted maximum likelihood method was used to calculate effect sizes. This offers a more conservative estimate of standard errors and is usually more sensitive to small samples [35]. Effect sizes were interpreted according to Cohen’s proposal [36], where *d* = 0.20 is small effect size, *d* = 0.50 corresponds to a medium magnitude, and *d* = 0.80 is considered a large effect.

The *Q* statistic of homogeneity [37] was used to assess the variability in the effect sizes of the studies. It was used to determine whether there was an unexplained variability in the selected studies [38]. The assumption of homogeneity is rejected when the *Q* statistic is statistically significant.

One-factor analysis of variance and meta-regression [39] analysis were used to perform meta-regressions to evaluate the 14 possible moderators of the effectiveness of self-affirmation interventions.

## 3. Results

### 3.1. Study Selection

The initial search returned 3819 articles. After identifying and removing duplicate studies from the database, the titles and abstracts of 2969 reports were reviewed and 2105 studies were excluded after the initial review. Subsequently, the full texts of 864 articles were reviewed, excluding 751 studies that did not meet the eligibility criteria. Finally, 114 research reports, comprising 144 studies, were included in the analysis. The flow diagram of the study selection process is shown in Figure 1.

### 3.2. Main Characteristics of the Included Studies

A total of 114 articles published between the years 1998 and 2023 were selected. Seventy-seven studies were conducted in the United States, whereas the remainder were conducted in the United Kingdom (n = 14), Canada (n = 6), the Netherlands (n = 5), China (n = 3), Germany, Belgium, Singapore, Spain, France, Australia, India, South Korea, and Turkey. All studies were authored in English. Regarding the samples, 61 studies considered samples with Caucasian participants, 11 with African American participants, and 13 with Latino participants. The predominant age range was 18–23 years (n = 108), followed by the age range of 11–14 years (n = 30) and 15–17 years (n = 5). A total of 36,419 students were included in the 114 studies. The experimental groups comprised 27,563 participants and the control groups (as reported) included 8856 participants. Regarding the dependent variables used to measure the effect of self-affirmation, 98 studies assessed psychological variables, 17 physical variables, and 50 performance-related variables. Table 2 summarizes the study details.

### 3.3. Assessment of Methodological Quality

Overall, the majority of studies exhibited a high methodological quality across all evaluated dimensions. The representativeness of the individuals selected in the studies was observed to be very strong. Additionally, most studies presented a moderate design quality and were categorized as analytical cohort studies (two pre + post groups). In terms of controlling for confounding factors, the majority of studies addressed at least 80% of the relevant factors (strong). Seventeen studies [23,41,45,46,49,50,51,55,57,59,61,68,109,121,124,128,134] indicated that evaluators and/or participants were aware of the research objectives (weak). Similarly, most studies employed validated and reliable data collection tools (strong). Regarding participant withdrawals and dropouts, the majority of studies described these aspects with moderate follow-up. These findings suggest a notable consistency in the methodological quality of the reviewed studies (See Figure 2).

### 3.4. Magnitude of Effectiveness

Specifically, our analysis of 144 studies with a total sample of 36.419 participants revealed a small and statistically significant effect size (*d*_IG+_ = 0.41, z = 16.01, *p* < 0.00), indicating that self-affirmation interventions can be effective in diverse sociocultural and educational contexts. The 95% confidence interval, which ranged from 0.36 to 0.45 (*SE* = 0.0253), supports the robustness of our results. Overall, the results suggest that self-affirmation interventions have the potential to produce positive changes in individuals’ attitudes and behaviors.

Homogeneity analyses revealed statistically significant variability or heterogeneity in the effect sizes. This suggests that the efficacy of self-affirmation interventions may be influenced by a variety of factors such as variations in study design, sample characteristics, and other methodological differences (*Q* (143) = 563.0122, *p* < 0.05, *v* = 0.06).

### 3.5. Moderators of Effectiveness

We explored several moderators associated with the sample characteristics and research design of different studies and found a large number of variables that influenced the variability of self-affirmation interventions. To assess and compare the percentages of variance accounted for by each identified moderator, we employed the coefficient of determination (*R*^2^) to ascertain the capacity of each moderator to explain the observed variability in effect sizes among the incorporated studies. These percentages provide the relative magnitudes of the distinct moderators’ influence (see Table 3).

The analysis revealed that, in interventions that target the Caucasian population (other), the impact of self-affirmation is likely to decrease by 7.07%. Further, the effects of studies applying self-affirmation to students aged 11–14 years (AR1) were 5.66% lower, whereas the effects of studies focusing on students aged 18–23 years (AR3) were 7.42% higher. Studies with small sample sizes (<500) showed a positive effect on the effectiveness of self-affirmation, whereas those with large sample sizes (>500) displayed a negative effect. This discrepancy suggests that, as the sample size increased, effect sizes tended to decrease by 7.72%. Concerning intervention group characteristics, in studies with a predominance of Caucasian participants (American, European, or Asian) and studies with predominantly ethnic minority students (African Americans, Latinos, etc.), the effects of self-affirmation decreased by 6.22% and 7.07%, respectively. In both cases, regardless of the participation of specific racial and ethnic groups, there was a decrease in the effectiveness of self-affirmation, indicating that ethnic and demographic characteristics may influence how these interventions affect outcomes. Finally, economic status had no significant influence on the effectiveness of self-affirmation interventions.

Regarding the moderators associated with research design characteristics, our results indicate that face-to-face interventions are more effective than virtual interventions, as the former can explain 5.26% of the data variability, underscoring the relevance of considering intervention designs. In addition, in studies that did not use software for random participant assignment, the effects of self-affirmation decreased by 4.33%, whereas in those that included such software, effects increased by 3.52%.

Concerning dependent variables, the results suggest that studies that focused on inducing changes in variables related to mental health (DV1) showed increased effects of self-affirmation by 12.08%, whereas studies aimed at improving academic performance (DV3) showed lower and negative effects (6.29%). These findings imply that self-affirmation interventions can have different effects depending on the outcome variable. Further, studies that included value writing as an intervention method found a negative impact on the effectiveness of self-affirmation interventions, explaining 7.87% of the variance. Typically, value writing as an intervention method is expected to have a positive impact on effect size; however, our results indicate otherwise. No significant differences in effect sizes were found in the studies that used a placebo activity in the control group.

Regarding the timing of self-affirmation interventions, regardless of whether the intervention was applied before or during potentially stressful events, the effects of the intervention were negative (7.79% and 3.70%, respectively). This suggests that interventions applied after such stressful events demonstrated a stability in effect sizes. Further, in studies in which teachers or educational assistants participated in the intervention, the effectiveness of self-affirmation decreased significantly by 4.63%. In other words, the presence of teachers or educational assistants appeared to be associated with a reduced positive impact of interventions. Finally, the way the studies were presented to students—whether as a routine academic activity or by informing them of the study’s objectives—did not significantly influence the variance in the effects of self-affirmation interventions.

### 3.6. Publication Biases

The results of this study showed that the findings were resistant to publication bias, as estimated by the Fail-Safe Number test for random effects meta-analyses [139]. The analysis suggested that 23.178 similar studies with null effect sizes would be necessary to nullify the results obtained in this study (see Table 4). However, the precision-effect test and precision-effect estimate with standard errors (PET-PEESE) meta-regression method was used following Staley and Doucouliagos’s [140] guidelines to further examine potential publication bias. The results demonstrated a presence of symmetry indices in the publication bias indicators, as the intercept of the PET estimate was not statistically significant, indicating that the constant in the PEESE regression could not adequately estimate the effect size values. Therefore, under these statistical parameters, the presence of publication bias in the study can be assumed (Table 5).

## 4. Discussion

This meta-analysis examined the efficacy of self-affirmation interventions in various educational contexts, considering demographic and methodological differences between the selected studies. A random effects model was used, and publication bias was assessed following Schmidt’s recommendations [141].

The results indicate that self-affirmation interventions are effective in improving academic performance, interpersonal performance, and aspects of physical and psychological well-being in academic contexts, obtaining an average effect size that is small but sufficient in view of the methodological differences between studies and the existence of moderators that manage to explain part of the variance. This suggests that the efficacy of self-affirmation interventions may vary across studies and in some cases a minimal or meager average effect size is observed, with the analysis of all moderators that may explain it being considerably important [32,33,142].

It is important to note that the literature recognizes predictable patterns that can explain the achievement of heterogeneous effects, such as temporal conditions of implementation, methodological procedures, social and cultural characteristics of the context, and individual differences of the subjects [32]. The findings of studies carried out in the last five years whose results show inconsistent, minimal, or even null effects of self-affirmation interventions present common explanatory assumptions related to individual differences between participants (coping with identity threat), the social and cultural context in which they operate, and differences in methodological procedures the between studies based on the provision of sufficient and necessary conditions for optimal fieldwork, taking as a reference the study by Cohen et al. [17], where the characteristics of the sample in terms of age of participants and size and degree of heterogeneity are usually critical points of disagreement between studies, in line with the results of the present meta-analysis [46,47,54,112,119,120,124,125,128,130].

In their meta-analysis, Wu et al. [143] found a significant moderation in the effect of a self-affirmation intervention by the achievement gap between threatened and nonthreatened students, duration of the study, presentation of the intervention as a classroom activity, and use of materials attached to those originally proposed. The present study complements these findings by adding, among others, the cultural background of the participants, their age ranges, the sample size, the characteristics of the dominant group in the sample, the modality of application of the intervention, and dependent variables such as mental health and academic performance. In addition, this meta-analysis included 144 independent studies, whereas the previous one included 58, which gives significant breadth to the scope of our conclusions. This is enhanced when considering that Wu et al. [143] used academic performance exclusively as the outcome variable, which was expanded in this work by considering additional outcome variables such as health and psychosocial variables. This represents an important contribution to the delimitation of interventions with a view toward enhancing their effects in different areas, which coincides with what has been proposed by other reviews, highlighting the investigation of moderating elements [32,33].

Regarding the moderators associated with sample characteristics, the effect of self-affirmation was weaker when the studies targeted students who were not threatened by stereotypes (Caucasian, European, or Asian). It appears plausible that students who did not encounter stigmatization or additional challenges in academic settings did not derive significant benefits from the interventions. A greater effect of these interventions on academic performance, interpersonal performance, and aspects of physical and psychological well-being is likely to occur when the perception of identity threat is greater [7,9,10,17,42,144,145,146,147,148,149]. These findings corroborate the results of previous studies that observed a reduced or null effect of self-affirmation among nonthreatened students [10,46,53,56,115].

The benefit of self-affirmation was greater when studies targeted individuals between the ages of 18 and 23, while it was lower in participants between the ages of 11 and 14. These differences in the effectiveness of self-affirmation may be associated with the stages of the students’ cognitive and socioemotional development. Young adults tend to have higher self-awareness and a greater ability to reflect on their values and beliefs, which could make the intervention more effective for them than for adolescents [150].

However, the results suggested that, as the sample size increased, the effect sizes of the interventions tended to decrease. In this regard, other studies suggest that the effects of self-affirmation could be more beneficial in social or cultural minorities and in small groups of students whose collective identity is underrepresented and who also experience learning difficulties, compared with the dominant group of students [20,32,151]. Therefore, it is plausible that threatened groups were included in studies with a small sample size (n < 500), which could potentially have influenced the effectiveness of the interventions. Further research is required to fully understand the interactions between these factors.

Similarly, the effect of self-affirmation was weaker in studies in which the sample was dominated by a specific racial group (e.g., Caucasians, African Americans, and Latinos). Recent research suggests that certain characteristics of social and cultural contexts, such as the size, composition, and distribution of groups, can become important sources of variability. For example, Bratter, Rowley, and Chukhray [6] found that identity threat in students of African American and Latin American background varied under certain group distribution conditions in educational settings. Although students of color belong to a socially and culturally marginalized group in the United States, they tend not to feel threatened when they are in the majority, rendering self-affirmation interventions ineffective. These findings suggest that studies with greater cultural diversity among participants may yield more consistent results regarding the effectiveness of interventions a predominant cultural or social group [44].

Regarding moderators related to research design, the results suggest that the effect of self-affirmation was greater when interventions were conducted face-to-face than virtual interventions. This finding aligns with the notion that face-to-face interactions can enhance emotional engagement and personal connections among participants, potentially amplifying the impact of self-affirmation interventions [152]. However, it is important to note that the effectiveness of virtual interventions can be influenced by various factors such as the quality of online platforms and participants’ levels of engagement. Future research should delve deeper into the mechanisms underlying this difference in effect sizes between intervention modalities to provide a more comprehensive understanding of their respective benefits and limitations.

However, the effects of self-affirmation interventions were greater when the studies did not use computer programs for random participant allocation. To some extent, these results contradict natural expectations. In general, one could assume that the use of rigorous randomization techniques would lead to more consistent and predictable intervention outcomes. However, there is also the possibility that the positive effects observed in these studies reflect a bias in sample selection, namely, a conscious or unconscious researcher bias during the process of assigning participants to the study groups [153]. These findings call for deeper and more thorough analysis to understand the potential reasons for this unexpected relationship.

The results indicate that self-affirmation had a more significant impact in studies that investigated its effect on dependent variables linked to mental health but had a less pronounced effect in studies that focused on measuring dependent variables related to academic performance. These results could be related to the very nature of self-affirmation, as it tends to strengthen self-esteem and a sense of personal worth—aspects that could positively influence individuals’ mental health [33,79]. Conversely, self-affirmation may not have a direct or immediate impact on academic skills or student performance. This could be because academic performance is influenced by a range of more complex factors, such as intrinsic motivation, socioeconomic status, the parents’ education level, and the educational environment [154,155].

Similarly, the benefits of self-affirmation diminish when studies include value writing as an intervention method for the experimental group. These results run counter to the expectations set by the literature, given that the application of alternative self-affirmation activities in the experimental groups should have generated these variations in effects, not the opposite [13,80,91,97,142,156,157]. This discrepancy between the expected and actual outcomes underscores the complexity of the factors that can influence intervention efficacy and highlights the importance of meticulously examining how different methods can impact results in specific contexts.

Similarly, the effect of self-affirmation diminished when interventions were administered before or during stressful academic events. These findings contradicted the literature, as some studies had found that the effects of self-affirmation tended to be more favorable when interventions are presented before stressful academic experiences associated with tasks or exams [9,11,151]. Each study had its particularities in terms of design, participants, and context, which could explain discrepancies in the findings. The way individuals perceive and cope with stress can also vary widely, which, in turn, could affect how they respond to self-affirmation during moments of academic stress.

The effect of self-affirmation was reduced when teachers or educational assistants participated in the intervention. These findings also differed from those previously made in the literature, which had shown that teacher support before and during academic activities can enhance the impact of self-affirmation [30,33]. The discrepancy between these findings suggests that the presence of teachers or educational assistants in the intervention may have had an unintended effect on students’ self-affirmation experiences. The dynamics of the interactions between teachers and students during the intervention may have played a crucial role in this effect. For instance, if teachers provided excessive direction or a rigid structure in the implementation of self-affirmation, students may have perceived the activity as less personal and authentic. This could have diminished the impact of self-affirmation interventions.

Some studies indicated that individual differences associated with the psychological impact of identity threats may be influenced by each subject’s personality traits. However, there is limited scientific evidence regarding this issue. It is important to recognize the relationships between individuals’ personalities and the way they perceive and evaluate themselves [158]. This evidence contributes to seriously considering personality features in the study of the effects of self-affirmation interventions, especially when mixed results are observed.

Other findings suggest that self-stigma for belonging to certain social or cultural groups, motivation and openness to improve or change, fear of academic failure, or even the relevance or meaning attributed to self-affirmation exercises may become essential moderators of the effectiveness of these interventions [33,101,159,160].

Taking into account all of the above, some guidelines for future studies whose interest is in investigating the effect of self-affirmation interventions in educational contexts are the choice of, ideally, small groups and culturally and socially diverse educational settings with different structures and systems of group organization.

The above sheds light on how identity threats are posed according to the size, composition, and distribution of the subjects, which few studies have ascertained. In the case of studies with large samples, it is suggested that an intervention proposal be implemented that includes not only self-affirmation as a strategy, but also others based on motivational factors. Further, pre- and post-intervention measures of identity threat should be incorporated in the groups to pre-identify individuals with higher levels of threat.

Future research should also explore the moderating effects of individual differences between subjects in the face of identity threats (e.g., personality traits or coping styles). Likewise, it would be a great contribution for future studies to explore what types of technical and methodological procedures are the most appropriate in terms of (1) the articulation of groups (experimental and control), (2) the characteristics of the self-affirmation or placebo activities, (3) the most suitable moments to execute the intervention, (4) the impact of the participation of members of staff (teachers, counselors, classroom assistants, etc.) before, during, and after the execution of the interventions, and finally, (5) the moderators associated with the contextual aspects already mentioned in previous paragraphs.

Regarding the limitations of this study, it is important to highlight that the results obtained show some evidence of publication bias. It is important to note that publication bias in a meta-analysis reveals a likelihood that the included studies are partially representative of the totality of studies conducted, but in the field of self-affirmation, the main reason for publication bias is the existence of more publications of studies reporting favorable results compared to those reporting unfavorable results. This phenomenon tends to occur because a study reporting evidence of the efficacy of self-affirmation interventions is more likely to be accepted for publication by a journal and therefore more likely to be cited and appear in databases when conducting the literature search required for a meta-analysis.

The publication bias of this meta-analysis highlights a problem facing the field of self-affirmation research, and these results may pave the way for enriching changes to make more room for the publication of studies reporting unfavorable results regarding the effectiveness of these interventions in an educational context.

Another limitation of this study is the generality of these interventions and moderators’ effect size estimates. This is because the calculation of effect size considered different types of outcome variables (psychosocial, academic, and health) to provide a broader view of the effectiveness of these interventions in educational contexts, despite the methodological differences between the studies, and to obtain results consistent with the objective proposed in the present meta-analysis. This also significantly enriches the decision-making of future researchers, whose interest is not limited to studying the effects of these interventions on students’ academic performance but also their impact on psychosocial or health aspects.

## 5. Conclusions

The results of the present meta-analysis suggest that self-affirmation interventions are effective and yield a small average effect size. Practically, these findings suggest that self-affirmation interventions may be useful tools in educational settings, particularly in terms of academic performance and student well-being. It is noteworthy that, owing to its low cost, this can be of particular benefit to schools in a risk context. This could be implemented through self-managed interventions designed by the educational institutions themselves and carried out by their own staff in accordance with established procedures to maximize their impact and enhance the resources of their students. From a theoretical perspective, according to our findings and the previous literature, it is important to emphasize the role of moderators in intervention effectiveness. The evidence suggests that with good delimitation and control for moderators, such as the age of participants, the intervention procedure, and the presence or absence of identity threats, the effectiveness of self-affirmation interventions could be maximized, which would have important implications at both the individual and educational system levels. Further research is needed to identify the most effective methods and moderators to optimize the benefits of self-affirmation interventions. In addition, specific analyses of the effectiveness of interventions considering individual independent variables would contribute, which is a limitation of the present study.

## Figures and Tables

**Figure 1 healthcare-12-00003-f001:**
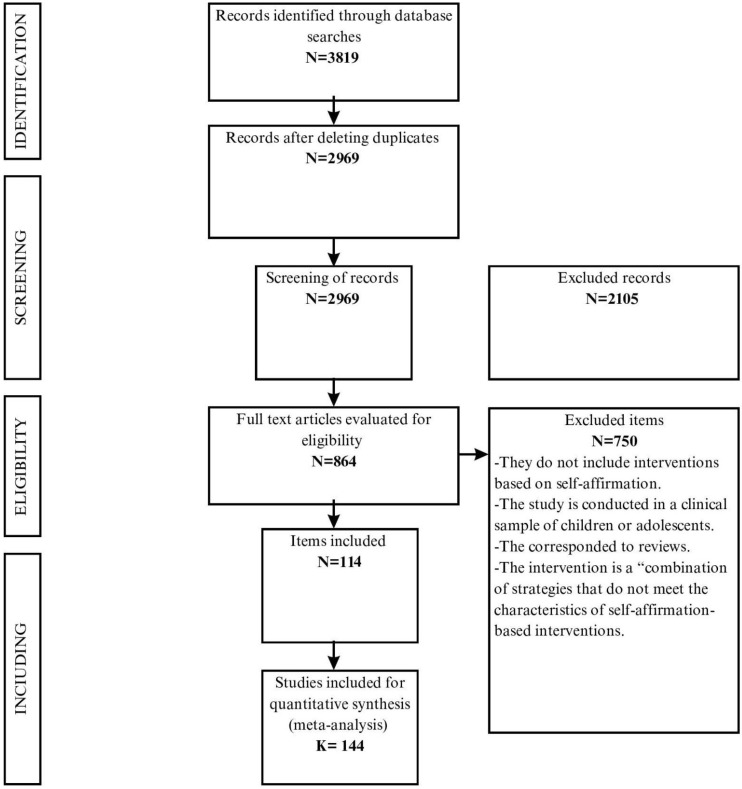
Flow diagram of the search strategy and selection of articles and studies.

**Figure 2 healthcare-12-00003-f002:**
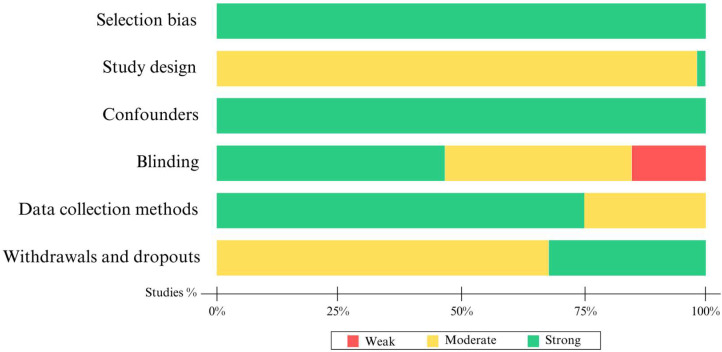
Summary of methodological quality of selected articles.

**Table 1 healthcare-12-00003-t001:** Potential moderators of the efficacy of self-affirmation interventions.

Moderators	Classification or Category
** *Sample characteristics* **	
Cultural origin	African American
	Latin American
	Ethnic minority
	Others
Age range	11 to 14 years old
	15 to 17 years old
	18 to 23 or older
Economic status	Medium
	Low
Sample size	Large sample (n > 500 students)
	Small sample (n < 500 students)
Dominant group in the sample	African American, Latin American, or ethnic minority
	American or European (Caucasian)
** *Design characteristics* **	
Execution modality	Face-to-face
	Virtual
Rigor of randomization	Use of software
	Does not use software
Dependent variable type	Psychological
	Physical
	Academic performance
Self-affirmation intervention style	Classical intervention (based on important values)
	No classical intervention
Placebo style	Classical activity (based on unimportant values)
	No classical activity
Time of the intervention	Before stressful academic event
	During stressful academic event
	After stressful academic event
Study presentation	Routine academic activity
	Activity for research
Teacher role	Active
	Passive

**Table 2 healthcare-12-00003-t002:** Characteristics and effect sizes of studies included in this meta-analysis.

Study(s) [Ref]	Year and Country	Sample Characteristics		Methodological Characteristics			Effect Size	
		Cultural Origin	Age Range	NC/NE	Variables	Results	Cohen’s d	Variance
Cohen et al. Study 1 [17]	2006, US	African Americans and European Americans	12–13	243	Academic performance	(+)	0.26 *	0.0166
Cohen et al. Study 2 [17]	2006, US	African Americans and European Americans	12–13	243	Academic performance	(+)	0.26 *	0.0106
Cohen et al. [8]	2009, US	African Americans and European Americans	12–14	385	Academic performance	(+)	0.31 *	0.0105
Bowen et al. [7]	2013, US	African Americans and ethnic minorities	11–14	74/58	Academic performance	(+)	0.52 *	0.0318
Thomaes et al. Study 1 [40]	2012, NL	Dutch	11–14	85/88	Prosocial feelings	(+)	0.47 *	0.0143
Thomaes et al. Study 2 [40]	2012, NL	Dutch	11–14	81/82	Prosocial behaviors	(+)	0.19 *	0.0156
Thomaes et al. [41]	2009, NL	Caucasians	12–15	405	Narcissistic aggression	(=)	0.10	0.0127
Armitage [42]	2012, GB	Caucasians	13–16	105/115	Perceived threat Self-esteem Current body shape Desired body shape Body satisfaction Body self-esteem	(−/+)	0.43 *	0.0182
Sherman et al. Study 1 [10]	2013, US	Caucasians and Latin Americans	11–14	92/92	Academic performance	(+)	0.33 *	0.0222
Sherman et al. Study 2 [10]	2013, US	Caucasians and Latin Americans	11–14	79/72	Academic performance Interpretation level Daily adversity	(+)	0.52 *	0.0207
Bratter et al. [6]	2016, US	African Americans, Caucasians, and Hispanics	14–15	456/430	Academic performance	(=)	0.09	0.0137
Cook et al. Study 1 [9]	2012, US	African Americans and Caucasians	12–14	361	Academic affiliation	(=)	0.16	0.0110
Cook et al. Study 2 [9]	2012, US	African Americans and Caucasians	12–14	121	Academic affiliation and academic performance	(+)	0.45 *	0.0343
Lokhande and Müller [43]	2019, DE	Ethnic minorities	12–13	294/374	Academic performance	(=)	0.14	0.0366
Binning et al. [44]	2019, US	Caucasians and ethnic groups	11–14	145	School trust Discipline incidents	(−/+)	0.40 *	0.0281
Hoffman and Kurtz-Costes [45]	2019, US	American Indians	11–14	212	Motivation for science	(=)	0.14	0.0189
Liu and Huang [46]	2019, CN	Asians	15–16	48/47	Self-integration Coping with homework Academic performance Perceived value	(−/+)	0.42 *	0.0423
Harackiewicz et al. [24]	2014, US	Caucasians and ethnic minorities	M = 19.27, SD = 1.15	396/402	Performance gap Academic performance	(−/+)	0.22 *	0.0263
Miyake et al. [25]	2010, US		18–22	399	Gender gap Learning	(=)	0.15 *	0.0216
Taylor and Walton [23]	2011, US	African Americans	18–22	29	Learning	(+)	0.83 *	0.0138
Baker et al. [47]	2020, US	Caucasians and ethnic groups		551/564	Academic performance	(=)	0.09	0.0036
Bayly and Bumpus [48]	2019, US	Ethnic minorities	18–19	107/389	Academic performance	(=)	0.00	0.0135
Blanton et al. [49]	2013, US		M = 18.7, SD = 1.16	116	Discomfort with the threat Willingness to have unprotected sex	(−/+)	0.43 *	0.0355
Brady et al. [31]	2016, US	Latinos	18–20	183	Academic performance Adaptive appropriateness Academic belongingness Spontaneous affirmation Fear of school Optimism Rumination Problem analysis	(+)	0.75 *	0.0204
Cameron et al. [50]	2015, GB	Caucasians and ethnic groups	16–24	799/696	Consumption of fruits and vegetables Physical activity Alcohol consumption Tobacco consumption Smoking in college	(−/+)	0.12 *	0.0286
Churchill et al. [51]	2018, GB		18–33	32/35	Musical performance	(=)	0.00	0.0598
De Clercq et al. [52]	2019, BE		18–19	129/123	Self-affirmation	(+)	0.47 *	0.0163
Covarrubias et al. Study 1 [53]	2016, US	Latinos	11–14	81	Academic performance	(+)	0.19 *	0.0106
Covarrubias et al. Study 2 [53]	2016, US	Latinos and Americans of European origin	11–14	269	Academic performance	(+)	0.22 *	0.0113
Ehret and Sherman [54]	2018, US	Caucasians and ethnic groups		74/66	Abstinence from alcohol	(=)	0.00	0.0356
Epton et al. [55]	2014, GB	Caucasians and ethnic groups	18–19	709/736	Tobacco use Drug use Hospital admissions Descriptive norms Perception of control	(−/+)	0.12 *	0.0199
Goyer et al. [56]	2017, US	Latinos and Caucasians	11–14	185	Academic preparation Attendance to selective schools	(+)	0.48 *	0.0227
Gregory et al. [57]	2017, US	Caucasian	18–22	64	Self-pity Perception of pain Resistance to pain	(+)	0.60 *	0.0139
Harackiewicz et al. [58]	2016, US	African Americans, Hispanics, and Native Americans	18–19	1040	Academic performance Performance gap	(+)	0.28 *	0.0254
Hernandez et al. [59]	2017, US	Latinos	11–14	67	Threat to identity	(=)	0.00	0.0615
Jones and Huey [60]	2020, US	Caucasians, Latinos, and African Americans	18–24	38/44	Academic performance Perception of self-integration Social adjustment	(=)	0.13	0.0130
Jordt et al. [61]	2017, US	Caucasians and ethnic minorities	18–22	970/963	Academic performance	(+)	0.27	0.0156
Kamboj et al. [62]	2016, GB		18–35	278/250	Pro-social feelings Alcohol consumption Intention to reduce consumption Message derogation Perceived threat Commitment to the threatening message	(−/+)	0.23 *	0.0109
Kim and Niederdeppe [63]	2016, SG	Caucasians and Asians	18–34	74/76	Negative cognitive responses Perceived risk of alcohol consumption	(=)	0.28	0.0267
Lannin et al. [64]	2013, US	European Americans and ethnic groups	19–46	84	Self-stigma Willingness to seek help	(−/+)	0.24 *	0.0627
Lannin et al. [65]	2020, US	European Americans and ethnic groups	18–22	152	Positive mood Negative mood Psychological distress	(−/+)	0.45 *	0.0295
Layous et al. [66]	2017, US	Caucasians and ethnic minorities	M = 19.12, SD = 1.28	57/48	Academic performance	(+)	0.39 *	0.0391
Meier et al. [67]	2015, US	Caucasians	18–35	52/58	Importance of the problem AR Risk perception AR Alcohol use Protective Strategies AR	(=)	0.16	0.0368
Norman and Wrona-Clarke [68]	2016, GB	Caucasians	M = 22.58, SD = 6.31	105/104	Reactivity of AR messages Message evaluation AR Perceived Risk AR Intention to binge drink Binge drinking	(=)	0.12	0.0192
Norman et al. [69]	2018, GB	Caucasians	M = 18.76, SD = 1.94	738	Frequency of excessive consumption AR	(+)	0.13 *	0.0054
Peters et al. [70]	2017, US	Caucasians and African Americans	17–59	194	Subjective numerical capacity	(=)	0.29	0.0208
Rosas et al. [71]	2017, US	Caucasians, Latinos, and ethnic minorities	18–35	143	Self-esteem Intention to consume sugar-sweetened beverages	(+)	0.24 *	0.0135
Sereno et al. [72]	2020, US	Caucasians, Latinos, and ethnic minorities	M = 20.04, SD = 2.69	157	Self-assessment Oral participation	(+)	0.45 *	0.0267
Shapiro et al. Study 3 [73]	2013, US	African Americans	18–24	37	Academic performance	(+)	0.19 *	0.0271
Shapiro et al. Study 4 [73]	2013, US	African Americans	18–24	75	Academic performance	(+)	0.57 *	0.0555
Tibbetts et al. Study 1 [74]	2016, US	Caucasians and ethnic minorities	18–24	69/72	Academic performance	(+)	0.23 *	0.0289
Tibbetts et al. Study 2 [74]	2016, US	Caucasians and ethnic minorities	18–24	389/399	Performance gap Choice of independent topics Choice of interdependent topics	(+)	0.29	0.0066
Walton et al. [75]	2015, CA	Caucasians, Asians, and ethnic minorities	18–24	228	Academic performance Importance of negative events Confidence in stress management Self-esteem Gender identification	(−/+)	0.60 *	0.0249
Adams et al. Study 1 [76]	2006, US	Caucasians, European Americans, and Latinos	18–24	44/51	Perception of racism Belief that whites understate the extent of racism Ratings of the average white person	(+)	0.58 *	0.0442
Adams et al. Study 2 [76]	2006, US	Caucasians, European Americans, and Latinos	18–24	27/36	Belief that whites understate the extent of racism	(−)	0.67 *	0.0688
Borman et al. [15]	2016, US	Caucasians and ethnic minorities	12–13	499/513	Cumulative seventh grade GPA Fall reading test Fall math test Spring reading test Spring math test Spring language usage test	(−/+)	0.05	0.0040
Briñol et al. Study 1 [77]	2007, ES		18–24	111	Manipulation check (index of personal importance)	(+)	2.51 *	0.0644
Briñol et al. Study 2 [77]	2007, ES		18–24	73	Manipulation check (index of personal importance)	(+)	1.84 *	0.0781
Briñol et al. Study 3 [77]	2007, ES		18–24	87	Attitudes	(−)	0.52 *	0.0475
Briñol et al. Study 4 [77]	2007, ES		18–24	91	Confidence	(+)	0.63 *	0.0461
Correll et al. [78]	2004, CA	Canadians	18–24	21/18	Advocate’s arguments Pro-attitudinal advocate position Argument strength	(−/+)	1.05 *	0.1182
Creswell et al. [79]	2013, US	Caucasians and ethnic minorities	18–34	73	Rating value Writing activity RAT score Positive affect	(+)	1.40 *	0.0582
Critcher et al. Study 1 [11]	2010, US		18–24	184	Defensiveness	(−/+)	0.15	0.0218
Critcher et al. Study 2a [11]	2010, US		18–24	76	Defensively negative score	(−)	0.57 *	0.0549
Critcher and Dunning Study 1 [29]	2015, US		18–24	75	Positive feelings of self-worth	(+)	0.45 *	0.0547
Critcher and Dunning Study 2 [29]	2015, US		18–24	94	Defensiveness Perspective on the threat	(−/+)	0.44 *	0.0435
Crocker et al. Study 1 [80]	2008, US	Caucasians, Asians, and other or mixed ethnicity	17–21	70/69	Rating of loving feelings	(+)	0.84 *	0.0319
Crocker et al. Study 2 [80]	2008, US	Caucasians, Asians, and other or mixed ethnicity	17–22	54	Acceptance of the article	(+)	0.63 *	0.1182
Dillard et al. [81]	2005, US	Caucasians	18–24	65/65	Motivated to quit smoking	(+)	0.39 *	0.0314
Epton and Harris [82]	2008, GB		18–46	41/46	Portions of fruit and vegetables Self-efficacy Response efficacy	(+)	0.46 *	0.0474
Harris and Napper [83]	2005, GB		18–24	42/40	Importance of self-affirmed passages Self-positivity Positive attitudes	(+)	5.08 *	0.4354
Harris et al. [84]	2007, GB	Caucasians	18–40	43/44	Threat Intention Control Self-efficacy Negative thoughts and feelings	(+)	0.67 *	0.0486
Klein et al. Study 1 [85]	2011, US		18–24	120	Feelings of vulnerability	(+)	0.36 *	0.0339
Klein et al. Study 2 [85]	2011, US		18–24	99	Feelings of vulnerability Intentions	(−/+)	0.43 *	0.0413
Klein and Harris [86]	2009, US		18–24	118	Attentional bias toward threat	(+)	0.37 *	0.0345
Koole and van Knippenberg [87]	2007, NL		18–24	88	Stereotypic word fragments Stereotypic descriptions	(−/+)	0.93 *	0.0511
Koole et al. Study 1 [88]	1999, NL		18–24	60	Value of the AVL subscale Recognition accuracy	(−/+)	1.49 *	0.0930
Koole et al. Study 2 [88]	1999, NL		18–24	71	Value of the AVL	(+)	1.84 *	0.0801
Koole et al. Study 3 [88]	1999, NL		18–24	70	Value of the AVL Recognition accuracy Positive mood Relative evaluation of name letters	(−/+)	0.99 *	0.0704
Legault et al. [89]	2012, CA		18–24	35	Errors of commission Waveform amplitude	(−/+)	0.82 *	0.1240
Martens et al. Study 2 [22]	2006, US		18–24	52	Items correct on math SATs	(+)	0.55 *	0.0799
Reed and Aspinwall [90]	1998, US	Caucasians, African Americans, Asian Americans, Biracials, Hispanics, and others	17–54	61	Reduced bias processing of threatening health information Reading time risk disconfirming Number of facts recalled Perceived control over reducing caffeine consumption	(−/+)	0.60 *	0.0688
Schimel et al. Study 1 [91]	2004, CA		18–24	49	Self-handicapping attributions Performance measures	(−/+)	0.57 *	0.085
Schmeichel and Martens Study 1 [92]	2005, US		18–24	65	Percept negative to anti-U.S. essay	(−)	0.50 *	0.0634
Schmeichel and Martens Study 2 [92]	2005, US		18–24	54	Death-related thoughts	(−)	0.54 *	0.0768
Schmeichel and Vohs Study 1 [93]	2009, US		18–24	63	Positive mood	(+)	0.44 *	0.0650
Schmeichel and Vohs Study 2 [93]	2009, US		18–24	72	Puzzle persistence	(+)	0.94 *	0.0617
Schmeichel and Vohs Study 3 [93]	2009, US		18–24	29	Behavioral descriptions	(+)	0.74 *	0.1474
Sherman et al. [94]	2009, US	Caucasians, Asians, Americans, other ethnicities	18–24	49	Concerns about failure Worrying during exam	(−)	−0.57 *	0.085
Sherman et al. Study 1 [95]	2000, US		18–24	60	Feel better about selves Point to most important value Accepting of threatening information Reduced caffeine consumption	(+)	1.14 *	0.0791
Sherman et al. Study 2 [95]	2000, US		18–24	61	Similar risk Considered risk of HIV Perceptions of risk	(+)	0.57 *	0.0683
Shrira and Martin Study 1 [96]	2005, US		18–24	101	Use of stereotypes Left hemisphere activation	(+)	0.50 *	0.0409
Shrira and Martin Study 2 [96]	2005, US		18–24	180	Left hemisphere activation Stereotyping	(−/+)	0.33 *	0.0226
Sivanathan et al. Study 2 [97]	2008, US		18–24	38	Reinvest funds in initially chosen Commitment to job candidate	(−)	0.94 *	0.1169
Sivanathan et al. Study 3 [97]	2008, US		18–24	55	Commitment to job candidate	(−/+)	0.61 *	0.0748
Spencer et al. Study 3 [98]	2001, US		18–24	24	Choice downward comparisons Choice upward comparisons	(−/+)	0.89 *	0.1840
Stone et al. Study 1 [99]	2011, US	Caucasians, Hispanics, Asians, and African Americans	18–24	179	Desire to meet target race	(+)	−0.30 *	0.0226
Stone et al. Study 2 [99]	2011, US	Caucasians, Hispanics, Asians, and African Americans	18–24	102	Desire to meet target race Empathy Guilt Perceived injustice Stereotyped views	(+)	−0.59 *	0.0413
van Koningsbruggen et al. [100]	2009, NL		18–24	84	Reaction time as a function of threat-related perceptions of message quality Reduction in caffeine consumption	(+)	0.47 *	0.0489
Vohs et al. Study 1 [101]	2013, US		18–31	52	Disengagement from a life goal	(+)	0.47 *	0.0571
Vohs et al. Study 2 [101]	2013, US		18–24	132	Performance expectations Dampening effect Interest in performing an additional task	(+)	0.36 *	0.0308
Vohs et al. Study 3 [101]	2013, US		18–24	119	More effort to task Dampening effect Effort to additional task Effort to attempt RAT problems	(−/+)	0.42 *	0.0344
Vohs et al. Study 4 [101]	2013, US		18–24	56	Negative self-perceptions of intelligence Self-perceptions of intelligence Self-efficacy perceptions Performance in the second set of RAT items	(−/+)	0.59 *	0.0746
Wakslak and Trope Study 1 [102]	2009, US		18–24	24	Self-concept clarity	(+)	0.92 *	0.1844
Wakslak and Trope Study 2 [102]	2009, US		18–24	45	Preferences for high-level action identifications	(+)	0.73 *	0.0948
Zárate and Garza Study 1 [103]	2002, US	Mexicans, Anglos, African Americans, and Asian Americans	18–24	120	Prejudices	(−)	0.36 *	0.0339
Zhao et al. [104]	2014, US	Caucasians, African Americans, Asians, Hispanics, and other races	18–24	116	Quitting intentions	(+)	0.29 *	0.0348
Sillero-Rejon et al. [105]	2018, GB		18–24	64/64	Avoidance Reactance Susceptibility Effectiveness Motivation to drink less Self-efficacy to drink less	(=)	0.18	0.0313
Pauketat et al. Study 1 [106]	2016, US		18–24	61	Affective forecasts Appraisal of negative event as less disturbing	(+)	1.02 *	0.0741
Pauketat et al. Study 2 [106]	2016, US		18–24	47	Affective forecasts Appraisal of negative event as less disturbing	(+)	.71 *	0.0905
Gu et al. [107]	2019, CN		18–24	48	Feedback-related negativity	(+)	0.82 *	0.0903
Taillandier-Schmitt et al. [108]	2012, FR		18–43	40/55	Performance scores Temporal performance scores	(+)	0.42 *	0.0441
Hanselman et al. [32]	2017, US	African Americans and Hispanics	12–14	166/165	Academic performance	(+)	0.24 *	0.0122
Borman et al. [109]	2018, US	Caucasians and African Americans	12–14	920	GPA	(+)	0.25 *	0.0044
Borman et al. [110]	2021, US	Caucasians and racial/ethnic groups	11–17	473/479	GPA	(+)	0.01	0.0042
Serra-Garcia et al. [111]	2020, US		18–24	283	Exam scores	(−)	−0.24 *	0.0142
Hayes et al. Study 1 [112]	2019, US	Latinos, African Americans, and Asian Americans	10–13	116	Overall semester grade	(=)	0.00	0.0345
Hayes et al. Study 2 [112]	2019, US	Caucasians, Latinos, and Africans Americans	18–22	273	GPA	(=)	0.00	0.0147
Hadden et al. [113]	2020, GB		11–14	562	Academic performance Levels of stress	(−/+)	0.21 *	0.0071
Scott et al. [114]	2013, AU		18–24	67/54	Intentions to reduce alcohol consumption	(+)	0.37 *	0.0340
Perry et al. [115]	2021, US	Caucasians and African Americans	20–43	416	Perceived residency competitiveness	(+)	0.20 *	0.0097
Dee [116]	2015, US	Caucasians, African Americans, and Hispanics	12–14	885	Grade in treated subject	(−/+)	0.30 *	0.0046
Protzko and Aronson [117]	2016, US	Caucasians, Hispanics and Africans Americans	13–15	243	Overall GPA	(=)	0.06	0.0165
Knight and Norman [118]	2016, GB	Caucasians	18–24	307	Self-affirmation manipulation	(+)	0.38 *	0.0133
Kim et al. [119]	2022, US	Hispanics and African Americans	9–10	29/37	Affect/emotion in relation to academic environments or tasks	(=)	0.27	0.0617
More et al. [120]	2022, US	Caucasians and ethnic minorities	18–28	125/129	Fear and defensive processing Exercise intentions	(=)	0.14	0.0158
Smith et al. [121]	2021, US	Caucasians and ethnic minorities	18–24	361	Task engagement	(+)	0.52 *	0.0115
Kim et al. Study 1 [122]	2022, US	Caucasians, African Americans, Asians, Hispanics, and others	M = 27.9	1277	GPA	(+)	0.07 *	0.0031
Pandey et al. [123]	2021, IN	Indians	22–27	40/40	Well-being	(+)	0.94 *	0.0709
Celeste et al. [124]	2021, GB	Afro-descendants and Caucasians	11–13	43/42	Cognitive performance	(=)	0.05	0.0476
Li et al. [125]	2022, CA, CN	Asians and Caucasians	M = 18.13, SD = 1.65	159/137	Psychological well-being	(=)	0.02	0.0136
Borman et al. [126]	2022, US	Caucasians, African Americans, Asians, Latins	12–14	2149	Suspensions	(−)	−0.28 *	0.0019
Binning et al. [127]	2021, US	Caucasians, African American, Latins, and Asian American	11–14	145	GPA	(+)	0.45 *	0.0283
Pilot and Stutts [128]	2023, US	Caucasians, African Americans, Asians, Hispanics, and others.	18–22	238	Body dissatisfaction Negative mood state	(=)	0.06	0.0168
Strachan et al. [129]	2020, CA	Caucasians and Asians	18–58	120	Exercise task self-efficacy	(+)	0.17 *	0.0339
Hagerman et al. [130]	2020, US	Caucasians, African Americans, Asians, Latinos	M = 19.35, SD = 1.61	167	Negative modo Absent-exempt Perceived skin damage Perceived vulnerability Intentions to protect skin	(=)	0.10	0.0241
Dutcher et al. [131]	2020, US		M = 19.3; SD = 1.35	27	Stressful Academic performance	(−/+)	1.24 *	0.1770
Shin et al. Study 2 [132]	2020, KR		M = 21.04, SD = 2.07	75	Acceptance of the threatening information	(−)	0.89 *	0.0586
Huppin and Malamuth [133]	2022, US	Asian American, European American, Hispanic American, African American, and others	18–24	70	Affirmative consent Conceptualization of consent Knowledge and awareness Rape beliefs Fairness of the outcome	(−/+)	0.61 *	0.0598
Çetinkaya et al. Study 1 [134]	2020, TR		M = 21.88, SD = 1.34	60	Task performance	(+)	0.84 *	0.0725
Poon et al. Study 4 [135]	2020, CN		M = 20.78, SD = 1.70	178	Conspiracy beliefs	(=)	0.83	0.0225
Turetsky et al. [136]	2020, US	Caucasians, African Americans, Asians, Hispanics, and others	18–44	108/118	Closeness centrality Degree centrality Maintaining existing friendships Forming new friendships	(+)	0.35 *	0.0193
Rapa et al. [137]	2020, US	Caucasians, African Americans, Asians, Hispanics, and others	M = 14.97	28/25	GPA	(+)	0.54	0.0785
Bosch [138]	2020, US	Caucasians, African Americans, Asians, Hispanics, and others	18–24	221/246	Academic performance	(=)	0.03	0.0086

Abbreviations: Number of subjects in the control group (NC); number of subjects in the experimental group (NE); mean associated with age (M); standard deviation associated with age (SD). Variable abbreviations: Related to alcohol intake (AR); Unadjusted grade-point-average (GPA); Remote Associates Test, a well-known measure of problem solving and creativity (RAT); Allport–Vernon–Lindzey (AVL); Force and Motion Concept Evaluation (FMCE). Abbreviations: Indicator of improvement in the dependent variable (+), mixed results with respect to the change in the dependent variable (−/+), absence of statistically significant changes in the dependent variable (=). Country abbreviations from https://www.nationsonline.org/oneworld/country_code_list.htm (accessed on 10 February 2023): Australia (AU), Belgium (BE), Canada (CA), China (CN), France (FR), Germany (DE), India (IN), Singapore (SG), South Korea (KR), Spain (ES), The Netherlands (NL), Turkey (TR), United Kingdom (GB), United States of America (US). * = *p* < 0.05.

**Table 3 healthcare-12-00003-t003:** Moderator variables and their impact on self-affirmation interventions.

Moderator	R^2^	β	Effect
** *Sample characteristics* **			
Cultural origin			
“Other” (Caucasian, European, or Asian)	7.07%	β = −0.21, z = −3.87, *p* < 0.001	(−)
Age Ranges (ARs)			
11–14 years old (AR1)	5.66%	β = −0.22, z = −3.45, *p* < 0.001	(−)
18–23 years old (AR3)	7.42%	β = 0.24, z = 3.98, *p* < 0.001	(+)
Sample Size			
Small sample (n > 500 students)	7.72%	β = 0.28, z = 4.04, *p* < 0.001	(+)
Large sample (n < 500 students)	7.72%	β = −0.28, z = −4.04, *p* < 0.001	(−)
Dominant Group in the Sample			
American, European, or Asian (Caucasian)	6.22%	β = −0.20, z = −3.59, *p* < 0.001	(−)
African American, Latino, or ethnic minority	7.07%	β = −0.18, z = −2.25, *p* < 0.05	(−)
** *Design characteristics* **			
Execution modality			
Face-to-face	5.26%	β = 0.26, z = 3.28, *p* < 0.05	(+)
Virtual	2.29%	β = −0.13, z = −2.14, *p* < 0.05	(−)
Rigor of Randomization			
Use of software	4.33%	β = −0.25, z = −2.97, *p* < 0.05	(−)
Does not use software	3.52%	β = 0.19, z = 2.66, *p* < 0.05	(+)
Dependent Variables			
Psychological (DV1)	12.08%	β = 0.28, z = 5.19, *p* < 0.001	(+)
Academic performance (DV3)	6.29%	β = −0.20, z = 3.59, *p* < 0.001	(−)
Self-affirmation intervention style			
Classical intervention (based on important values)	7.87%	β = −0.27, z = −4.06, *p* < 0.001	(−)
Timing of Intervention			
Before stressful academic event	7.79%	β = −0.22, z = −4.04, *p* < 0.001	(−)
During stressful academic events	3.70%	β = −0.23, z = −2.81, *p* < 0.05	(−)
Teacher role			
Active	4.63%	β = −0.20, z = −3.10, *p* < 0.05	(−)

Abbreviations of the results: Indicator of improvement in the dependent variable (+), indicator of decrease in the dependent variable (−).

**Table 4 healthcare-12-00003-t004:** Fail-Safe Number test for a meta-analysis with random effects [139].

Fail Safe Number	
Rosenthal (0.050)	43.646
Rosenberg Normal	32.963
Rosenberg t-N1/t-N+	22.790/23.178 (three iterations)

**Table 5 healthcare-12-00003-t005:** PET-PEESE meta-regression indicators based on the Staley and Doucouliagos [140] approach to reduce publication selection bias.

Sample	PET		PEESE	
	β_0_	β_1_	β_0_	β_1_
Full	−0.03 (−0.11, 0.05)	2.57 **	0.14 ** (0.09, 0.18)	7.96 **

For PET-PEESE, β0: Constant or intercept in the meta-regression; β1: Coefficient of standard error in PET meta-regression and variance in PEESE. Values in parentheses correspond to the lower and upper limits of the 95% confidence intervals. ** = *p*-value < 0.01.

## Data Availability

All data are contained in the article tables.

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
