# Peer review of "Effectiveness of Self-Affirmation Interventions in Educational Settings: A Meta-Analysis"

_healthcare, 2023, doi:10.3390/healthcare12010003_

Round 1
Reviewer 1 Report
Comments and Suggestions for Authors
Respected Authors
The authors conducted a meta-analysis to assess the effectiveness of self-affirmation intervention in educational settings. Your manuscript is of interest but needs some revisions as follows.
- Title, in my opinion, "Self-Affirmation" is a better choice than "Self-Affirming". If approved, please modify it here and through the text.
- Abstract, please remove all headings from your abstract.
- Line 17, meta-analytic evidence is not a good term here. There are a few differences between meta-analytic evidence and meta-analysis. Please refine it here and through the text.
- Lines 19-20, this statement needs modification. This statement is related to the results section. Please remove it here. Also, it is better to use the word "included". For example, "Finally, 144 experimental studies were included in this study".
- Abstract, please add the exact date of search. Also, add brief information regarding quality appraisal and meta-analysis approach.
- Abstract, please add more results in this section.
- Please revise your methods and results considering items of the PRISMA 2020. Also, please attach a filled PRISMA checklist as a supplementary file.
- Based on PRISMA and other guidelines, the first section for all systematic reviews and meta-analysis in the method is "eligibility criteria". Please, revise it.
- In my opinion, your manuscript needs to be modified by a native editor. There is several punctuation, grammatical errors, and improper correction throughout the text.
- Line 184, replace "descriptors" with "keywords" or "search terms".
- Please add a full search strategy for all databases including the date of search and the number of search records as a supplementary file.
- Lines 198-199 are not related to methods. Please remove it here and add it to the results section.
- Lines 213-215, remove from methods. These numbers are not matched. 113 articles met the inclusion criteria and then, 144 studies were included in the meta-analysis. how is it possible? Also, please remove all Figures and Tables from the methods section. All related to the results.
- Methods section, please add clear information regarding study selection, quality appraisal, and data extraction considering the PRISMA checklist.
- Results, your results are interesting but restructure this section considering the PRISMA checklist.
- Please present some of your mata-analysis results using Figures and Tables. It is not recommended to mention all results in the text. First, it is not recommended to present the results in this way, and second, it makes it difficult for the reader to understand.
-
-
Comments on the Quality of English LanguageIn my opinion, your manuscript needs to be modified by a native editor. There is several punctuation, grammatical errors, and improper correction throughout the text.
Author Response
Dear Reviewer
We are very grateful for the time and effort you have invested in reviewing our article. Please find below the changes and corrections we have made, which you can review in detail in the attached document.
Please see attached file

Reviewer 2 Report
Comments and Suggestions for Authors
This paper sought to understand how self-affirmation interventions affect students in educational contexts, drawing from a huge number of studies and considering a range of factors that might influence the outcomes. Here are some concerns of this study.
First, the introduction section is comprehensive and covers broad aspects related to self-affirmation. However, there are evident repetitions and redundancies, particularly in describing the role of self-affirmation and its implications in educational settings. It might be beneficial to restructure the section to reduce redundancy and enhance clarity.
Second, the method section doesn't specify how articles were screened. The screening process typically involves title and abstract screening followed by full-text assessments. Additionally, coding and data extraction are unclear. There is no mention of how the data from the selected studies were coded or who did the coding.
Third, while the moderators have been extensively listed, it's not clear if the selection was based on prior theoretical or empirical grounds. It is suggested to justify the choice of each moderator based on previous literature.
The discussion section is well-researched and comprehensive. However, while the section mentions publication bias, it quickly dismisses its impact, stating it "does not detract from the importance or merit of the results." This is a bold claim made by authors without any cited sources, as publication bias can significantly skew results. There should be cited sources to support this claim if the authors insist.
Author Response

(The authors gave the same response as above.)

Reviewer 3 Report
Comments and Suggestions for Authors
The manuscript entitled “Effectiveness of Self-Affirming Interventions in Educational Settings: A Meta-Analysis” is an interesting and valuable addition to the education literature. It concerns a meta-analysis of the impact of self-affirming interventions in educational settings. In spite of methodological differences among the selected studies, the evidence points to a reliable beneficial effect of this type of intervention. In my modest opinion, the article merits publication if a few concerns are addressed.
The content of the abstract may need to be simplified. The authors may consider writing shorter sentences.
“Meritorious mean effect size” sounds awkward. Shall you select a different adjective? Focusing on the size of the effect may be sufficient.
The authors may consider the conceptual difference between a threat approach and an opportunity approach in addressing self-affirming interventions. Individual differences exist in the way the same educational setting is approached by students. See
Vandewalle, D., Nerstad, C. G., & Dysvik, A. (2019). Goal orientation: A review of the miles traveled and the miles to go. Annual Review of Organizational Psychology and Organizational Behavior, 6, 115-144. https://doi.org/10.1146/annurev-orgpsych-041015-062547
If I were the authors, I would bear in mind that the dependent measures used by the selected studies vary considerably. The authors state that they selected studies that “report the effectiveness of this intervention on academic performance, aspects of psychological well-being or interpersonal development, offering intra- and intergroup comparisons”. How can the authors justify their combining studies with qualitatively different dependent measures? The scales of measurements are also different.
The conclusion section of the manuscript leaves the reader asking for an account of the implications and applications of the results of the meta-analysis conducted by the authors. Broadly speaking, what are the practical implications? What are the theoretical implications of these results?
The entire manuscript may benefit from proofreading to ensure that the intended meaning is successfully conveyed. Also, the rules of capitalization may be reviewed.
Comments on the Quality of English LanguageExtensive editing of the English language is required.
Author Response

(The authors gave the same response as above.)

Round 2
Reviewer 1 Report
Comments and Suggestions for Authors
Respected Authors
Thank you for your modification.
Acceptable changes have been made in the article but still, some things need to be corrected or not changed well. For example, based on comment number 5, I recommended that you follow the PRISMA checklist but you have followed this checklist to some extent and the reason you mentioned is not correct. Even in the process of conducting a systematic review, it is necessary to determine eligibility criteria before searching for articles. Please correct it. Also, line 219 still has the previous error and the incorrect word "meta-analytical" is used.
What do you mean by the "codification process"? I have never seen such an item in a systematic review. With the explanations mentioned below, I think "data extraction and data items" will be more suitable.
Also, the information related to the evaluation of the quality of the articles is missing both in the method and the results section. This section is mandatory in systematic review studies. Please add the desired information.
Cheers
Author Response
Dear reviewer
We are very grateful for the time and effort you have invested in reviewing our article. Below you will find the latest changes and corrections we have made, which can be reviewed in detail in the attached document.
Please check the attached file.

Reviewer 2 Report
Comments and Suggestions for Authors
This paper presents a comprehensive and methodologically sound investigation into an important area of educational psychology. I just have two concerns:
1. While limitations such as publication bias are mentioned, a more thorough discussion on how these limitations might impact the findings is needed.
2. While the paper concludes with practical implications in the conclusion part, how could these findings be practically applied? Concrete examples or recommendations would be valuable.
Author Response

(The authors gave the same response as above.)

Reviewer 3 Report
Comments and Suggestions for Authors
The authors have adequately addressed the concerns expressed in the earlier review of the manuscript.
In the abstract, I would change the last three sentences. The authors may consider the following text to clarify the implications of the results of their meta-analysis:
"Through a meta-analysis of the impact of self-affirmation interventions in educational contexts, this study suggests that interventions are effective, resulting in a small mean effect size. Thus, self-affirmation interventions can be considered useful, brief, and inexpensive strategies to improve general well-being and performance in educational settings."
Comments on the Quality of English Language
Moderate editing of the English language is required. Several sentences are awkward.
Author Response

(The authors gave the same response as above.)

Round 3
Reviewer 1 Report
Comments and Suggestions for Authors
Respected authors
Thank you for your clarification.
Cheers
Reviewer 2 Report
Comments and Suggestions for Authors
This paper presents a comprehensive and methodologically sound investigation into an important area of educational psychology. The authors have addressed all the concerns raised in the previous reviews. The discussion section now more accurately reflects the implications of the findings, which enhances the overall contribution of the work to the field. The manuscript now meets the publication standards and I have no further suggestions for revision.